# Two Theorists on Work Systems: Murray Bowen and Barry Oshry

Katherine Kott

Institute for Social Innovation, Fielding Graduate University, Santa Barbara, CA 93105, USA;
kkott@email.fielding.edu

**Abstract:** This paper explores the development of two theories of human behavior as they relate to work systems. Both Murray Bowen and Barry Oshry formulated theories of how people operate in groups. Bowen developed his theory through observation of families and extended his thinking to apply more broadly. Oshry observed work systems in his lab and thought what he saw there could also be true in families and society at large. Practitioners have applied both theories in their work with groups. However, neither theory has received much attention in terms of the theoretical concepts they contain or the processes the theoreticians used to generate them. The purpose of this paper is to examine the methods these two theorists use to create their frameworks, compare and contrast the theories they posited as a result, and consider the possible future development for them.

**Keywords:** Bowen theory; Oshry; systems theory; work systems

## 1. Two Theorists on Work Systems: Murray Bowen and Barry Oshry

Apparently independently of each other, Murray Bowen and Barry Oshry developed theories of human behavior based in systems thinking during the 20th century. Each of these systems thinkers came upon their ideas from different fields of study and focused on different groups. Neither intended to become theorists but both were presented with the opportunity to do so when they tried to make sense of what they observed. They had similar ideas about what constitutes a theory; not simply a good idea but a "number of interlocking concepts" [1] (p. 305) that are descriptive and predictive and open to further development [2].

Bowen, a psychiatrist, developed his theory by studying the families of schizophrenic patients, while Oshry's work originated in large-scale organizational models built to support undergraduate business education. In 1946, Bowen began a psychiatric residency at the Menninger Foundation in Topeka, Kansas. From there, he moved to the National Institute of Mental Health (NIMH) where he launched an experimental project to study schizophrenia within the context of the family. Initially, Bowen planned to use Freudian theory as the principle for organizing his findings but ultimately, he found it impossible to fit his observations into the Freudian model. As Rakow explained, the Freudian model focused on "pathology within the individual" [3] (p. 113) attributed to problematic parenting. Treatment of the individual outside of the family context could overcome the individual's early experience. Bowen began to see the family as a resource rather than a problem, however. "If one believed that nature determined the family to be the best location of health for a child, then strengthening the parent's capabilities, without blame, makes sense for the generations to follow" [3] (p. 113). Freudian theory could not be extended to fit Bowen's beliefs about the human as part of nature and what he was observing in families. Therefore, he was inspired to create a new theory of human behavior. From NIMH, Bowen moved to the Georgetown University Medical Center in 1959 [4] (p. 371). In the process of setting up the program at Georgetown, Bowen became interested in what he called "administrative systems" [4] (p. 373). Based on his experience in the organization that later became the Bowen Center, Bowen recorded his ideas about the theoretical concepts that apply to work systems [5]

Because the only portion of his work that was set up as an experiment was the NIMH project, the methodology Bowen used to generate his theory is not documented in the same way an effort to create a theory in an academic setting might be. Therefore, no statistics about the number of families or organizations he studied to formulate his theory have been compiled. Nonetheless, it is evident that his ideas began to emerge at NIMH. In addition to the three families enrolled in his study, Bowen was able to observe several families under the care of other psychiatrists who were housed in the same ward. Rakow [3] documented Bowen's awareness of the emotional process in the work system during the NIMH project. Referring to the expectations of staff on the project, Rakow stated, "A lack of concealment and working on self were early precursors to an understanding of a better functioning society. The notion of staff paying attention to self was a point of differentiation in an emotional system. Change in one member influenced change in another" [3] (p. 214).

Bowen refined and tested his theoretical concepts as he coached hundreds of families and individuals with less severe problems than schizophrenia. Andrea Schara interviewed more than forty people who studied with Bowen or were coached by him. The Murray Bowen Archives Project posted recordings of these oral history interviews on their website [6]. This interview data provides a glimpse into the ways Bowen put his theory to the test and is a subset of the information Bowen gathered from the hundreds of families and individuals he treated and coached (A. Schara, personal communication, 20 February 2023).

Oshry's plan in establishing the Power Lab in 1970 was to create an experiential learning environment in management education. In the 1960s, prior to starting the Power Lab, he "developed large scale organizational simulations for undergraduates in business" [7] (p. 271) at Boston University where he was a faculty member. He also did research and led workshops at the National Training Laboratories (NTL) during that decade. "At the time, I was less focused on theory development than on creating engaging interactive experiences" [8] (p. 547). Yet, as he analyzed the data that were collected through observation of the simulations in the lab, processes that were difficult to describe or explain through conventional focus on individual personality traits began to emerge. Similarly to Bowen, Oshry was then motivated to develop a framework to accurately depict what he saw.

As an emergent theory rather than a tested hypothesis, Oshry's work also lacks the typical methodological data associated with an academic study. However, in acknowledging that he learned and devised his theory by observing his students, he gave an indication of the scope of his work. "I am indebted to the many thousands of people who have participated in our Power Labs and Organization Workshops and who have allowed me to be with them, observe them, and interview them as they wrestled with the challenges of system life" [7] (p. xxvi). Table 1 summarizes the theorists' backgrounds and what drove each of them to develop a theory.

Despite the difference in their origins, the resulting frameworks have their overarching systems approaches in common and are either implicitly or explicitly based on the way natural systems operate. Both theorists also thought it was important to base their theories on the observation of human behavior as a living system rather than on reports from people about their actions. "The back and forth interaction between the parts and the whole is observable and predictable in living systems. Systems thinking focuses on the facts of how the parts of a network interact and under what conditions the patterns of interaction change" [9] (p. 1).

Bowen developed his theory based on the observation of families in the 1950s and by the late 1970s, he had extended some of its concepts to apply to work groups and society. Even before he had completed work on his family systems theory at NIMH, Bowen made note of "the process in the group" of his staff at Menninger [3] (p. 133). His most focused effort to see systems at work was through the observation of his own behavior "as Head of the Family Faculty and the Family Programs at Georgetown University" in the early 1970s [10] (p. 461). However, in a teaching video made in 1979, Bowen also commented on the functioning of two individuals he thought exhibited good leadership, one as a team leader or department head at the State Department and one as a school superintendent.

Several people who knew Bowen at that time thought perhaps these individuals were clients whose work lives he had learned something about in the therapeutic process. On occasion, Bowen was also called upon to evaluate the mental health of a person in a work system but as far as anyone could remember, Bowen did not consult to organizations.

**Table 1.** The Theorists.

| Bowen | Oshry |
| --- | --- |
| Psychiatrist | Business school professor |
| Studied families with schizophrenic members | Established experiential learning environments for members of organizations |
| Active from 1946 until his death in 1989 | Active from the 1960s until the present (at age 90) |
| Sought treatment for schizophrenia | Sought experience for students that would expose them to power dynamics in organizations |
| Observed "emotional process" in families | Observed the consequences of hierarchy in laboratory simulations |
| Saw the need for a new natural systems theory of human behavior (not Freudian) | Saw the need for a natural systems theory based on hierarchy |
| Extended theoretical principles from family to human systems, including work groups (administrative systems) | Speculated that concepts might apply to other groups beyond work groups (political, social, family) |

After Bowen's death in 1990, colleagues at the Georgetown Family Center, now the Bowen Center, continued to explore the application of Bowen theory to work systems. *The Emotional Side of Organizations* [11] was published in 1995 and the Bowen Center currently offers an annual course: *Differentiation at Work* for students of Bowen theory who are interested in its application to work systems. Unfortunately, the Bowen Center is not set up to offer credentials such as certificates, course credits, or degrees other than CEUs for mental health professionals. While this restriction ensures that course participants are serious in their study of Bowen theory, it discourages individuals and organizations that value credentials as part of the continuing education process from participating in research based on Bowen theory.

Bowen theory has been referred to in various ways over the years since Bowen developed it, most notably as Bowen Family Systems Theory or BFST. However, the organization Bowen launched, now called The Bowen Center for the Study of the Family, and the satellite organizations affiliated with the center simply refer to the theory as Bowen theory, making room for the application of the theory to groups other than families.

Building on the organizational models he had created for business education in the 1960s, Oshry studied work groups in his lab and described what he saw there. Later, he speculated that the same mechanisms that he observed in work groups would be present in families and other human systems. Oshry remains affiliated as founder of Power + Systems, the consulting firm he and his wife and partner Karen founded in the 1970s. His theory has also been referred to in various ways including "leading systems" from the title of his 1999 book [12] (p. 79) although Oshry himself refers to it primarily as a "theory of whole systems" [13] (p. 12) or an "organic systems framework" [2] rather than by name. In this paper, I will simply refer to his theory as Oshry's theory.

Although both Bowen theory and Oshry's theory have received attention from practitioners who could see the value each framework brings to work groups [12,14], neither theory has seen in-depth analysis of the concepts each theorist proposed in this context. For Bowen theory, this gap has occurred for some structural reasons, including isolation from the typical scholarly communication process [15] and has stood in the way of advancing Bowen theory, even as it applies to family businesses. Bowen was so caught up in the practical application of his theory that he often avoided spending time writing up his research for formal publication, although he did deliver a number of papers as he formulated his ideas. Along with an extensive collection of videotaped teaching lectures and interviews, the published versions of these conference papers make up the primary Bowen theory canon.

One incentive for writing this paper came from the frustration experienced by a researcher who was interested in using Bowen theory to understand succession in family-owned businesses. His doctoral committee refused to consider Bowen theory. As he explained:

"[A] critical element for uptake is openness to challenge and modification of BFST via thorough comparative research. All theories can be continuously fine-tuned and tested using the latest or more established research methods. Many PhD supervisors would be reluctant to support the use of research that is more than 5 years old, let alone 30+, and [that] hasn't been reviewed or used by recognized academics in the field. (P. Wilde, personal communication, 15 May 2022)."

It is less clear why Oshry's theory has not seen wider consideration as a theoretical framework as well as a practical methodology for improving work group performance, although Oshry acknowledged that he did not write and publish an academic article describing his theory until 2020 [2], decades after he developed it. Perhaps it is also because, as both Bowen and Oshry have expressed, it is difficult for people to "see systems" rather than simpler causal thinking. As Bowen stated, "Man is deeply fixed in cause-and-effect thinking in all areas that have to do with himself and society" [16] (p. 420). In addition, Oshry and Bowen exhibited communication styles and impatience with others that may have discouraged academic attention. In describing Oshry's critique of the way his program was handled at NTL in *The Systems Letter,* Mirvis said, "the story meanders across time and in-and-out of the narrator's self-conscious and animated mind" [17] (p. iv); not the type of treatment one would expect to find in academic discourse.

Due to the lack of academic treatment for these two theories, it is challenging to situate them in the organizational systems field. Based on natural systems, both theories focus on what might be considered biological processes rather than organizational structure or operational practices. It is possible that some systems thinkers were influenced by Bowen or Oshry but did not credit them because they were unable to find published material that described the theories. Perhaps Bowen and Oshry can be thought of as lost ancestors. Hirschhorn and Gilmore [18] did cite Bowen's triangle concept in a paper on the application of structural family therapy to organizational behavior. However, Hirschhorn situates his current practice within psychodynamics, rooted in Freudian theory, rather than family systems theory [19]. Cambridge Leadership Associates' core principles [20] seem to owe something to Bowen theory, although they do not acknowledge its influence. In particular, they mention the importance of recognizing behavioral patterns, which Oshry also emphasized. Jessup noted that Oshry saw the need to be aware of one's own perspective within the system as well as to develop the skills of an outside observer [12] (p. 80). Heifetz, Linsky (of Cambridge Leadership Associates), and Grashow encouraged leaders to develop outside observer capacity through an exercise they called "On the Balcony" [21] (p. 9).

Both Bowen and Oshry observed human behavior closely and saw the complexity of interrelationships, despite their shortcomings in describing what they saw for academic audiences. The concepts they developed have some similarities and some differences. Each theorist observed work group processes and drew conclusions about the importance of understanding human behavior in work groups as systems processes. Some of their observations could be considered complementary. Both are worthy of more consideration than they have received in the literature. The goal of this paper is to create an initial platform through which both Bowen theory and Oshry's theory can gain wider exposure as theoretical frameworks that apply to work systems in an academic forum and thereby provide an opportunity for future theoretical development of these approaches.

## 2. Bowen Theory

Kerr and Bowen [4] described the natural systems theory that Bowen developed through observation of families as it pertains to families and family therapy in detail, and the Bowen Center's website also covers the seven concepts Bowen formulated with families in mind. The seven concepts are triangles, differentiation of self, nuclear family emotional

process, family projection process, multigenerational transmission process, emotional cutoff, and sibling position. In addition, Bowen later added the concept of societal emotional process. It offers a template for applying the theory to non-family groups [22]. In this paper, I will focus on describing and explaining concepts Bowen identified as most pertinent to work groups, directing the reader who is interested in learning more about the remaining concepts to works by Kerr, the Bowen Center, and others cited there.

In their summary of Bowen theory concepts, Papero et al. [9] focused on how the ebb and flow of anxiety and stress generate interactive patterns of behavior between and among individuals. Bowen called this emotional process. The emotional process in family systems is generated by pressure from two opposing forces; the drive to be an autonomous individual and the need to be part of a group. It is important to note, as Kerr reminded us, that Bowen thought that anxiety in humans was similar to any other organism's "reaction to real or imagined threat" [23] (p. 108).

In writing about what he called work and administrative systems, Bowen emphasized the importance of acting based on principles and beliefs as an autonomous self, what he called differentiation of self [5,10]. He also discussed the mechanism of interactions he called triangles, whereby when tension between two people becomes intolerable, one of the members of the dyad brings in a third person. "[A] relationship problem can arise between two people and transmit itself through interlocking triangles to employees at lower levels" [5] (p. xi).

Bowen often presented his thinking in less formal settings. In two undated interviews with Kathleen Wiseman, Bowen [24] responded to her request for him to describe a well-functioning leader by discussing the idea that the "average person" believes that leadership is positional within a hierarchy. Bowen went on to say that he "tried to use the idea of responsible self" in thinking about leadership. A leader is a person who is responsible for the self and "certain important others." Regardless of the positional hierarchy, a person can demonstrate leadership from any level of the organization. That said, the administrative leaders of an organization set the tone and their behavior can have consequences that reverberate throughout the organization. Bowen provided an example of this happening within the National Institute of Health (NIH) when difficulty at the top level of the organization seemed to cascade down to the lowest levels of the organization. In the case he described, rather than being willing to see their own part in the process, the top leader identified a low-level researcher as a problem employee [25]. Despite the fact that Bowen did not label what he saw happening as such, it seems reasonable to think that Bowen might have seen this process as projection, similar to the projection process he had observed in families. In the family projection process, rather than keeping a problem contained within the marriage, the parents project the family tension onto one or more of the children. The idea that Bowen thought projection occurred in work systems as well as family systems is upheld by his statement that "[t]he basic patterns in social and work relationships are identical to relationship patterns in the family, except in intensity" [10] (p. 462). Nonetheless, when Bowen focused on work systems in his writing, he identified differentiation of self and triangles as the most pertinent concepts [5] (pp. x–xi).

### 3. Oshry's Theory

Oshry situated his theory in a fundamentally hierarchical framework consisting of what he called "systemic relationships—Top/Bottom, End/Middle/End, and Provider/Customer" [13] (p. 13). This framework grew out of an experimental learning community called Power & Systems Laboratory set up in 1972 [26] (Introduction). In a laboratory setting, Oshry generated his theory by observing people's behavior in a simulated work system. The simulation consisted of an organization that had customers and potential customers. Participants played the roles of directors/tops, managers/middles, workers/bottoms within the organization, and customers, outside of the organization [13] (p. 12).

The emphasis on hierarchy and the associated power dynamics grew in a landscape where few pioneers in organization development were considering these factors in early

laboratory education models such as T-groups. While T-groups emphasized personal growth, critics such as Bennis believed that developing an understanding of systems and power dynamics was missing from such training modalities [17] (p. vii). Apparently, the notion that organizational systems operated on the basis of power dynamics resonated with Oshry. He developed a hierarchical model and observed participant behavior to test his ideas about "patterns of systemic relationships" and "patterns of systemic processes" [13] (p. 13).

The top group was seen as the group primarily responsible for the success of the operation and could fall into a pattern of burdened over-functioning. The bottom group, if not held accountable, could become irresponsible and under-function. As a result, leaders responsible for long-term strategy might be incapacitated by being caught up instead in less important but seemingly urgent issues that should be handled at a lower level of the organization [13] (p. 15). Similar incapacity can develop when those in the middle are torn between meeting the expectations of those above them in the hierarchy by extracting something from the people they manage and responding to the complaints of their subordinates. In Oshry's view, those in the middle tend to be put in competition with each other and develop a sense of disconnected powerlessness as a consequence [13] (p. 28).

## 4. Similarities and Differences

The fundamental similarity between these two theories is their foundation in natural systems. In addition, both theorists formulated their theories through observation. Table 2 lists the similarities in their approaches and beliefs.

**Table 2.** Similarities.

| |
| --- |
| Natural systems framework |
| Emphasizes systems process over individual personality traits |
| Based on observation |
| Belief that understanding the process could improve functioning |
| Belief that it is possible to move out of fixed position (of feeling oppressed) |
| Belief that work systems functioning can be improved |

As a psychiatrist interested in the treatment of people diagnosed with schizophrenia, Bowen observed families in a clinical setting at the NIMH from 1954 to 1959, although the seeds for new thinking had been sown during his years at the Menninger Foundation from 1946 to 1954. Bowen credited the leaders at the Menninger Foundation for cultivating an environment that encouraged creative thinking. "They were more interested in helping young people develop their own capacities than in communicating a fixed body of knowledge" [4] (p. 349). In his early work with schizophrenic patients, Bowen focused on the symbiosis between the patient and the mother. However, he observed that what he came to call the emotional process involved family members beyond the dyad and shifted his focus to the family system, bringing entire families to live on the research ward. "Each family included the two parents, one maximally impaired schizophrenic offspring, and one or two normal children" [4] (p. 361).

As Bowen strove to develop a systems framework to correct difficulties he saw with Freud's emphasis on the individual, he considered other systems theories that had been developed in the mid-twentieth century. Rather than adopt either Ludwig von Bertalanffy's or Norbert Wiener's theory, both of which he thought were too mathematical, Bowen settled on a framework that was "designed to fit precisely with the principles of evolution and the human as an evolutionary being" [4] (p. 360). Von Bertalanffy was a biologist best known for the development of a general systems theory. Although his theory applied to living things, he used mathematics to model an organism's growth [27]. Bowen did not believe mathematical models fit a scientific approach to human behavior. Without a direct connection to biology, Norbert Wiener, a mathematician and philosopher, was perhaps even more removed from the natural systems framework Bowen sought to develop [28].

"There was general systems theory, which was then developed [by] Bertalanffly, [sic] which included concepts from mathematics. I had thrown out mathematics because it was

based more on the way the human thinks than on what the human is. That would be a way of making it a non-science. There was another way of thinking which was based on mathematics, according to principles developed by Herbert [sic] Weiner . . . I didn't want either one. I developed a natural systems theory based on the notion of man as a passenger on the planet. It went back to the beginning of evolution. An estimated 4000 million years ago. So that would be the difference in systems theories" [24] (p. 3).

Oshry observed work groups in a laboratory setting to confirm his hypothesis that "We human beings are the most social of social creatures . . . [y]et we are blind to the workings of the whole systems of which we are a part" [13] (p. 12). Similarly to Bowen, Oshry broadened his initial focus, which was on power dynamics in organizations, to a systems view by observing what actually went on in the groups [2].

Although Bowen and Oshry studied different populations, they reached some of the same conclusions about human behavior. In fact, Bowen's observation, that the patterns he observed in the families with a schizophrenic member were present in all human groups, put his project at risk as the funding received was intended to support research specific to schizophrenia, not generalized observations on human behavior [4] (p. 367).

Both Bowen and Oshry emphasized the importance of a research or neutral perspective in developing the ability to see systems [7] (p. xiv) or hear descriptions of how they operate described [4] (p. 349). Both theorists also believed that an increased understanding of the processes they identified could shift "dysfunctional systemic relationships" [13] (p. 31). Taking responsibility for oneself, especially as a leader, could create the conditions for a "real happy crowd" [29].

Whereas Bowen saw an overall emotional process at work in any group that was closely affiliated, be it a family or work system, Oshry saw distinct processes within and between each hierarchical level. Bowen did not place the same emphasis on hierarchy that Oshry did but he did acknowledge the role hierarchy plays in work systems. He described developing awareness of his own tendency to be "overresponsible" [10] (p. 463) as what Oshry would call a "Top," Ref. [13] (p. 13) taking on the work of others and actually making it harder for staff members to take responsibility for their own portfolios. Table 3 summarizes the differences between the two approaches.

**Table 3.** Differences.

| Bowen | Oshry |
|---|---|
| Emotional process<br><br>- Anxiety<br>- Triangles | Hierarchy<br><br>- Tops, middles, bottoms |
| Differentiation—acting on principle | Differentiation–system response to threats/opportunities—one of four patterns that also include:<br><br>- Individuation<br>- Integration<br>- Homogenization |
| Individual leadership within the system | Group action within the system |

Both theorists believed it possible for people at lower levels of the hierarchy to get out of fixed positions characterized by feelings of oppression [13] (p. 15) and complaints. While a good leader would set clear expectations, employees could request better definition of their responsibilities if they were unclear. Bowen believed that in most cases, a responsible lower-level employee would be able to resolve difficulties by getting clarification. In the few cases where the leader was unable or unwilling to provide direction, the employee could choose to leave rather than continue complaining [29].

Oshry thought such requests were likely to be more complex as the employee would be trying to change a relationship pattern. "Your effort to change from one side of the relationship

is likely to perturbate the party or parties on the other side" [13] (p. 16). He recommended "a conversation about how each of you is experiencing the current condition: what shared vision you two have for the success of this project or process and how changing the pattern could strengthen your relationship and the work you are engaged in" [13] (p. 16).

This difference in thinking about the best way for an individual to approach a work system issue characterizes the major contrast between the approaches offered by each of the two theorists. While Bowen emphasized understanding the system while taking individual responsibility within it, Oshry identified the greatest potential for change in group action within the system. The concepts each theorist used to describe work system processes and the opportunities for improved functioning further distinguish the two approaches. Each theorist also developed specific terminology, sometimes using the same terms to mean different things.

For Bowen, the fundamental opportunity for improved functioning in work systems as well as in the family is through "differentiation of self" [5] (p. x). A differentiated person is one who acts based on "principle instead of feelings and subjectivity" [5] (p. xi). In Oshry's model, differentiation refers to the entire system and describes structural, process, and strategic response to threats and opportunities [13] (p. 18). In natural systems terms, differentiation, as Oshry describes it, might be thought of as adaptation. Differentiation is one of four patterns that comprise the relationships and processes in a system. The other three patterns in Oshry's theory are individuation, integration, and homogenization. When a system individuates, the parts operate independently. When it integrates, the parts work collaboratively; when it homogenizes, the whole maintains commonality across the system. In Oshry's model, all four patterns need to be at work and in balance for the "whole organic entity" to function optimally [13] (p. 17). Oshry also included the need to balance power and love within the operation of these four patterns. He adopted this idea from Adam Kahane's work [13] (p. 18). It is interesting to think about power and love as two opposing forces and somewhat confusing to think about how these two forces fit in with the patterns Oshry observed in the lab.

While Bowen shied away from the use of the term individuation, he did discuss two opposing forces: individuality and togetherness. In this pairing, individuality is related to the concept of differentiation, the ability to act thoughtfully based on principles, whereas togetherness pulls individuals to go along with a group.

"Bowen proposed that individuals . . . continuously respond to two powerful instincts or forces. The first is to be an emotionally autonomous individual, free from the constraints of relationships to pursue one's own goals and plans. The second is to be connected to others and a part of the group" [9] (p. 4).

This idea of togetherness is different from Oshry's concepts of integration and homogenization as it is a component of an "emotional system" [4] (p. 10), instinctive processes that involve "people's encroaching on one another and functioning at the other's expense" [4] (p. 94). In Bowen's framework, it is possible to achieve group cohesion without individual thoughts and ideas being lost in the process. He believed this was possible through responsible leadership.

Bowen chose to use ordinary terminology as labels for his concepts. However, his use of terms was not always consistent with ordinary dictionary definitions or commonly understood meanings. For example, within the Bowen theory framework, emotion is understood not as a feeling but as a biological response to stimulus, perhaps more akin to instinct. Thus, an emotional system or process consists of interactions that are driven by "stress and emotional tension in various relationships" [9] (p. 4). Bowen called this driver anxiety. Unlike Oshry's use of love as a force that balances or opposes power, Bowen, presumably in an attempt to adhere closely to describing human systems as part of nature, did not use terms such as love, associated with feeling states.

Oshry identified several drivers that cause systems to respond "reflexively" rather than thoughtfully. When a system is challenged by complexity, it reacts by differentiating, whereas if it is threatened, it responds by integrating. When a system is torn, the system

reacts by dispersing through individuation [13] (pp. 19–20). Both Bowen and Oshry thought these automatic patterns could be interrupted through awareness and a redirection of attention from assignment of dysfunction to individuals to the processes at work.

One of the observations that stimulated Bowen to shift his focus from the dyadic symbiosis between the mother and schizophrenic child was the movement of anxiety beyond the dyad. He noticed that when the tension between two people exceeded the relationship's capacity to contain it, one of the people would bring in a third person, forming a triangle.

"[T]he triangle, a three-person emotional configuration, is the molecule of the basic building block of any emotional system, whether it is in the family or any other group. The triangle is the smallest stable relationship system. A two-person system may be stable as long as it is calm, but when anxiety increases, it immediately involves the most vulnerable other person to become a triangle. When tension in the triangle is too great for the threesome, it involves others to become a series of interlocking triangles" [30] (p. 373).

Bowen listed interlocking triangles as one of the concepts of his theory most applicable to work systems. "In emotional systems such as an office staff, the tensions between the two highest administrators can be triangled and retriangled until conflict is acted out between two who are low in the administrative hierarchy" [31] (p. 175).

Oshry's theory does not include triangles as an overall mechanism. However, the process of triangling, as Bowen described it, can be seen in Oshry's middles at both the individual and the group level. Oshry stated, "Middleness is the condition in which we exist between two or more individuals or groups; these groups have differing priorities, perspective, goals, needs, and wants; and each of them exerts pressure on us to function on its behalf" [26] (location 76). In Oshry's model, this condition is embedded in the middle position of the hierarchy, whereas Bowen saw the possibility that anyone could be put under pressure to function on behalf of another person through the mechanism of a triangle.

Just as Bowen thought it possible to "de-triangle" [4] (p. 151) through a differentiating move, Oshry provided an example of how an individual, Daniel, got out of the middle position at work by arranging a meeting between two departments rather than acting as the go-between as he had done in the past [26] (location 398). However, Daniel was warned not to try such a move again in the future, a reaction Bowen called "change back" [23] (p. 86). Anyone attempting to make a change in the emotional process of a system should expect and be prepared for this to occur.

Both theories have the advantage of describing and promoting a broader view of work systems to enable seeing dysfunctional patterns for what they are rather than focusing on individual failings (scapegoating).

"Right now, the default reaction to tension or breakdowns in relationship is to experience them *personally*. The problem is you and your temperament, character, or personality defect or me and my temperament, character, or personality defect, or possibly, an incompatible mix of our two different orientations" [13] (p. 15).

Both theorists sought a way to move beyond this rigid focus on individual behavior to a more holistic and flexible way of seeing humans in relation to one another in the workplace. Despite the differences in their frameworks, each theory offers opportunities for possible improvements in workplace systems function, perhaps using different mechanisms.

Bowen theory offers the opportunity for individuals to make a difference. Oshry's theory might require attention from a consultant or coach. Using the example of Daniel, who made a differentiating move, in Bowen theory terms, or worked to get out of the "tearing" [13] (p. 26) middle position in Oshry's theory, how might each theorist envision a way for Daniel to determine whether or not his new stance was sustainable? Bowen theory might suggest that Daniel would need to decide whether or not he could continue to work under the expectations set by his boss, or if there was any possibility his boss might become more flexible if Daniel stood his ground. Oshry might focus instead on an intervention that would involve Daniel, his boss, and his subordinates to raise awareness of the patterns they had fallen into due to their hierarchical positions as tops, middles, or bottoms. They might

also look for imbalances in the four elements that Oshry believed must operate together: individuation, integration, differentiation, and homogenization [13] (p. 17).

## 5. Opportunities for Future Development

Murray Bowen and Barry Oshry were motivated to become theorists by observing human systems. Bowen developed his theory by observing families on a psychiatric ward and in his clinical practice. He extended his theory to work systems primarily through observation of his own work systems. Oshry developed his theory by observing simulated hierarchical work systems in a laboratory setting. While practitioners who have been exposed to the two theories have applied them in their work as managers and consultants, limited exploration of the theories in peer-reviewed academic writing has restricted the evaluation of these two frameworks for understanding work systems. This paper is intended to be a platform from which future study of Bowen theory and Oshry's theory can be made.

It would be interesting and valuable to test Oshry's theory by observing a work system rather than observing a simulated work environment in a laboratory setting. What might be particularly interesting would be to study a new work system such as a start-up to see if the structures that Oshry defined emerge. Perhaps both frameworks could be tested in the same work setting.

Paul Wilde, quoted at the beginning of the paper, was interested in using the Bowen theory to study succession in family businesses but was stymied by the lack of recent published information about Bowen theory as it applies to work systems. With its roots in the study of the family, Bowen theory would be an ideal framework to use to evaluate leadership transitions in family businesses. What difference does it make for family leaders to be working on differentiation of self as a new leader is selected and responsibility for the firm moves from one generation to the next? Is it possible to observe triangles in the process?

If, as both Bowen and Oshry believed, a theory is not simply a good idea but related concepts that describe and predict behavior, it should be possible to test the validity of their descriptions by continuing to observe work systems. In addition, students and researchers who are interested in human behavior and work systems can build upon the ideas these two theorists generated to correct, extend, and update these theories. Does hierarchy always emerge with the tensions Oshry described within and between the groups? Are his descriptions and labels of the four components required for a system to be robust accurate? When a system is anxious, is it possible to see the emotional system through triangles as Bowen did? Can a leader who sets clear expectations calm the system down as Bowen predicted? Exploring these research questions could deepen the understanding of human behavior in work systems Bowen and Oshry have offered.

**Funding:** This research received no external funding.

**Data Availability Statement:** No new data were created or analyzed in this study. Data sharing is not applicable to this article.

**Acknowledgments:** The author gratefully acknowledges the Institute for Social Innovation at Fielding Graduate University for the fellowship award that provided library access for this project. She also appreciates time and useful suggestions from the reviewers. In addition, members of the Bowen theory community have generously contributed to the content of this paper. Staff of the National Library of Medicine, History of Medicine Division supported research in the Bowen archive there. Any errors or omissions are however, mine alone.

**Conflicts of Interest:** The author declares no conflict of interest.

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
