# Peer review of "Two Theorists on Work Systems: Murray Bowen and Barry Oshry"

_systems, doi:10.3390/systems11030138_

Round 1
Reviewer 1 Report
Two theories of human behavior in a group are reviewed. Their common and differing issues are outlined. Some directions to further study are proposed. The text is written clearly and well understandable.
The drawback of the paper is lack of formal structures. I think it is advisable to arrange the discussed matter (differences in the theories, similarities in the theories, steps of development of the theories et c.) in form of tables and/or charts rather than only as a bulk text.
Also it would be interesting to know more about practical basis of these theories, including numbers. How many families were treated by Bowen himself to draw his theory? How many families were treated afterwards? Same for Osher.
References 16 and 20 are to Wikipedia. I think this is not acceptable since Wikipedia can be modified by virtually anyone and anytime. References to works of Bertalanffy and Wiener themselves or other research papers can be given.
Line 258: Herbert -> Norbert;
Line 295: unnecessary linebreak.
Author Response
Many thanks for your thoughtful comments. My indented responses follow each point below.
The drawback of the paper is lack of formal structures. I think it is advisable to arrange the discussed matter (differences in the theories, similarities in the theories, steps of development of the theories et c.) in form of tables and/or charts rather than only as a bulk text.
Thank you for this suggestion. I agree that it will make the comparisons easier to see with the addition of some tables. I will make this change.
Also it would be interesting to know more about practical basis of these theories, including numbers. How many families were treated by Bowen himself to draw his theory? How many families were treated afterwards? Same for Osher.
This is also a good suggestion. It would be easy to get information from Catherine Rakow's recent book about how many families Bowen observed at NIMH to originate his theory. However, I think information about how many families he treated and how many individuals he coached as he solidified his theory would be anecdotal. I'll consider the suggestion and see if I can gather some information that I think is reliable. Compiling such data would be an interesting follow-up research project. Perhaps the data on Oshry will be easier to obtain from the consulting firm. I'll look into that as well.
References 16 and 20 are to Wikipedia. I think this is not acceptable since Wikipedia can be modified by virtually anyone and anytime. References to works of Bertalanffy and Wiener themselves or other research papers can be given.
Although I think more people are accepting Wikipedia as a valid source, your point is well taken. I will swap in more standard sources at this point in the paper.
Line 258: Herbert -> Norbert;
Oops! Thanks for catching this. I will make this correction.
Line 295: unnecessary linebreak.
This happened in the transfer from APA format to MDPI format. I'll fix this and also check the other spots where indented quotes in APA format may need quotation marks added in MDPI.
Reviewer 2 Report
The review is attached

Author Response
Thank you for your thoughtful comments. I agree that the paper would be strengthened with the addition of tables and will add those.
I believe I can add some context, as you suggest, to situate the two theorists within the field of organizational systems. However, to add detailed information about other theories and comparisons with them would be an entirely different paper. The purpose of the paper, as you observe in your first paragraph, is to shine a light on two people who have been largely ignored in the field. I think a more general survey of other theories would shift the focus away from the two lesser known theorists and create more of a review article. I will plan to enrich the context but not include a large amount of additional material on other theories.
Round 2
Reviewer 1 Report
The author has filled the gaps in the text. The paper can be published.
Reviewer 2 Report
No more comments